# Psychometric and Structural Validity of the Pittsburgh Sleep Quality Index among Filipino Domestic Workers

**DOI:** 10.3390/ijerph17145219

**Published:** 2020-07-20

**Authors:** Peng Xiong, Adam P. Spira, Brian J. Hall

**Affiliations:** 1Division of Medical Psychology and Behavioral Sciences, Department of Public Health and Preventive Medicine, School of Medicine, Jinan University, Guangzhou 510632, China; paulxiongwhu@gmail.com; 2Global and Community Mental Health Research Group, Department of Psychology, The University of Macau, Macau (SAR) 999078, China; 3Department of Mental Health, Johns Hopkins Bloomberg School of Public Health, Department of Psychiatry and Behavioral Sciences, Johns Hopkins School of Medicine, Johns Hopkins Center on Aging and Health, Baltimore, MD 21205, USA; aspira@jhu.edu; 4Department of Health, Behavior and Society, Johns Hopkins Bloomberg School of Public Health, Baltimore, MD 21205, USA

**Keywords:** PSQI, validity, CFA, EFA, Filipina domestic workers

## Abstract

*Objectives*: Evaluate the psychometric properties and structural validity of the Filipino version of the Pittsburgh Sleep Quality Index (PSQI) among Filipino domestic workers (FDWs). *Methods*: In Study 1, 131 FDWs completed PSQI and other scales, along with 10-day actigraphic assessment with accompanying electronic daily sleep dairy. A subsample of 61 participants completed follow-up assessment after 10 days. In Study 2, 1363 FDWs were recruited and randomized into two halves. Exploratory factor analysis (EFA) and Confirmatory factor analysis (CFA) were used in the two halves, respectively. *Results*: In Study 1, the Cronbach’s alpha of the PSQI was 0.63 at baseline and 0.67 at follow-up. Test-retest reliability for the PSQI global score based on intraclass correlation was 0.63. Convergent validity was supported by the significant associations between the PSQI global score, PSQI components scores, sleep patterns from the daily sleep diary, and measures of depression, anxiety, and rumination. Small correlations between the PSQI global score and measures of daytime sleepiness, social support, and self-reported height, supported discriminant validity. In Study 2, EFA yielded two PSQI factors with acceptable factor loadings. CFA established that this two-factor model, comprised of perceived sleep quality and sleep efficiency, evidenced better model fit than alternative models tested. The Cronbach’s alpha of two factors was 0.70 and 0.81, respectively. *Conclusions*: The PSQI demonstrated good internal consistency of two factors, and good convergent, and divergent validity. Results can be referenced in future studies to measure and screen sleep dysfunction among clinical and non-clinical populations in the Philippines.

## 1. Background

The healthcare-related burden of impaired sleep is enormous. Studies increasingly link inadequate sleep and sleep disorders like insomnia to increased risk of depression, and other mood disorders [1,2,3], as well as increased fatigue [4], reduced psychomotor performance [5], poor memory consolidation [6], and substantial workplace cost due to work underperformance and absenteeism [7].

Migrant workers are likely to experience increased risk of poor sleep and consequent poor health. Migrant workers, especially domestic workers, may be exposed to sleep deprivation due to on-call nature of their work, even during nighttime hours, exposure to traumatic stress, worries about family separation, and long working hours in general [8,9,10,11]. However, the literature on migrant worker sleep problems is scarce. One cross-sectional study about transnational Latino migrant farmworkers revealed that 11% reported daytime sleepiness [12]. In a prevalence study of sleep problems among asylum seekers and refugees, more than half of them (75.5%) reported moderate to severe sleep disturbance [13]. The absence of data on the prevalence of sleep-related problems among transnational migrants is a significant gap in the sleep literature. 

Currently, there are 2.3 million Filipino domestic workers (FDWs) across the world [14]. One previous study with 32 migrant Filipino live-in caregivers (average age = 51 years old) in the US reported average low sleep quality (3.3 out of 5 based on a five-point sleep quality questionnaire) along with excessive daytime sleepiness (40.0%) [15]. Evidence from other studies also suggests that labor migrants experience insomnia, sleep deprivation, and poor sleep quality, accompanied by high levels of anxiety and depression [16,17]. Based on the updated 2018 by the Labour Affairs Bureau, Macao Special Administration Region (SAR), there are 27,348 non-resident domestic workers in Macao. Among them, 52.06% (14,238) come from the Philippines (http://www.dsal.gov.mo). The principle reason for migration from the Philippines is to seek better economic circumstances and improved financial support for their families [18]. Several studies conducted in Macao found a high (>25%) prevalence of anxiety and depression [19]. posttraumatic stress disorder [11], and a 5% prevalence of gambling problems [20], and discrimination was associated with these disorders [21]. All of these disorders can exacerbate sleep problems.

A study of 290 FDWs in Hong Kong measured their mental health symptoms and revealed that they were suffering much from loneliness, worry, lack of social support, and sleeping difficulties [22]. However, no previous study has attempted to validate a sleep measure among the large Filipino migrant labor force. Quantifying the burden of sleep impairment requires valid and reliable scales. 

The Pittsburgh Sleep Quality Index (PSQI) is one of the most widely used self-report sleep measures and has 19 items designed to evaluate subjective sleep quality over the past month [23]. The questionnaire contains seven component scores that range from 0 to 3: ‘subjective sleep quality,’ ‘sleep latency,’ ‘sleep duration,’ ‘habitual sleep efficiency,’ ‘sleep disturbances,’ ‘use of sleeping medication,’ and ‘daytime dysfunction.’ The seven component scores are summed to obtain a global score ranging from 0–21. Based on the original study, scores larger than 5 indicate poor sleep quality, which yielded a specificity of 86.5% and a sensitivity of 89.6% in distinguishing good and poor sleepers [23]. Higher scores on each component indicate poorer sleep. This scale has been translated into many different languages and is a well-established scale with acceptable psychometric properties among numerous clinical and non-clinical populations [24,25,26]. 

Although the PSQI was originally introduced as an instrument with a unidimensional factor structure [23], studies debated whether two- or three-factor model better represents impaired sleep than a single factor model [27,28,29,30]. A summary of previous factor structure studies are summarized in Table 1. Taken together, there is no clear evidence of a best fitting model for the PSQI. 

We performed two studies to evaluate the psychometric and structural validity of PSQI. Study 1 aimed to evaluate the psychometric properties of the PSQI among FDWs. In study 2, the primary aim was to examine the factor structure of the PSQI using exploratory and confirmatory factor analytic methods in this population. The secondary aim was to compare the relative fit of the alternative one-, two-, or three-factor structural models of the PSQI found in previous studies. 

## 2. Methods

### 2.1. Participants

In Study 1, a total of 131 female Filipino domestic workers (FDWs) were recruited using snowball sampling from March 2016 to September 2016 in Macao (SAR), People’s Republic of China. All participants answered the baseline questionnaires, and 61 of them were invited to complete the PSQI once again after 10 days. The data of this study was a part of a larger study utilizing actigraphy to determine the burden of sleep dysfunction and related correlates, along with several embedded validation studies [11,19,21] and a pilot study for the larger planned respondent driven sampling (RDS) project [20,31,32,33].

In Study 2, the data with 1363 FDWs was obtained from a RDS project conducted in Macao (SAR) from November 2016 to November 2017. 

The studies were approved by the ethics committee of the University of Macau. The research process and objectives were explained to the participants before the informed consent was acquired.

### 2.2. Measures

In Study 1, the Filipino versions of the PSQI and Epworth Sleep Scale (ESS) were provided by the Mapi Research Trust (https://eprovide.mapi-trust.org). Official translated versions of the PHQ-9 and GAD-7 were obtained from Pfizer [34]. The Ruminative Response Scale (RRS) and Multi-Dimensional Scale of Perceived Social Support (MSPSS) were translated into Filipino following standard forward and backwards translation guidelines, including cognitive interviews, and pilot testing [35]. Actigraphy and daily sleep diaries were used in Study 1. In Study 2, only PSQI questionnaire data and demographic information were used.

Objective Sleep:

The Actiwatch-2 (Philips Respironics, Bend, OR, USA) is a widely used wrist-worn sleep-monitoring device, validated against PSG, and used to monitor sleep patterns and individual sleep quality [36]. All the participants wore the actigraph on the wrist of their non-dominant hand for 10 continuous days with 30 seconds epoch length. We only used data from eight nights, removing weekend nights, which reflect different sleep patterns. The following outcome variables are generated: total sleep time (TST), sleep onset latency (SL); sleep efficiency (SE); wake after sleep onset (WASO); number of wake bouts (WB); and fragmentation index (FI), which is an indication of the degree of sleep fragmentation (detailed in Table 2). 

Daily Sleep Diary

This consisted of self-reported TST, bedtime and wake time, SL (assessed on a 5-point ordinal item ranging from ‘less than 15 minutes’ to ‘more than 120 minutes’), sleep quality (SQ) (assessed on a 5-point ordinal item ranging from ‘very good’ to ‘very bad’), TIB (the total time spent in bed), and SE (detailed in Table 2). The diary records of bedtime and wake time were also used to clean the sleep logs in Actiwatch-2. Daily sleep diary was received via online survey sent using short message service twice per day (morning and evening).

Depressive Symptoms

The Patient Health Questionnaire with nine items (PHQ-9) is a self-report screening measure used to assess depressive symptoms occurring in the past two weeks. Each item is rated from 0 (not at all) to 3 (nearly every day). Higher total scores indicate greater depression symptom severity [37]. The Filipino version of PHQ-9 was used in a previous study among FDWs in Macao with a good internal consistency (Cronbach’s alpha = 0.79) [38], and validity [19,39]. The Cronbach’s alpha in the present study was 0.78 indicating good internal consistency reliability. 

Anxiety

The Generalized Anxiety Disorder scale with seven items (GAD-7) was used to measure anxiety symptoms [40]. Each item is rated from 0 (not at all) to 3 (nearly every day), with an anxiety symptom severity score from 0 to 21. The Filipino version of GAD-7 was used in the previous study among FDWs in Macao with a good internal consistency (Cronbach’s Alpha = 0.80) [38], and validity [19]. The Cronbach’s alpha in the present study was 0.82 indicating good internal reliability. 

Epworth Sleepiness Scale:

The Epworth Sleepiness Scale (ESS) is an 8-item self-report questionnaire to measure daytime sleepiness in adults [41,42]. Items range from (0 ‘never’ to 3 ‘high chance’) to reflect subjects’ probability of falling asleep in eight different situations (e.g., while sitting or reading, watching television, and driving). The total score of ESS ranges from 0 to 24, with higher scores indicating greater daytime sleepiness [41]. The Cronbach’s alpha in the present study was 0.82, indicating good internal reliability.

Rumination

The Ruminative Response Scale (RRS) short version describes rumination that is self-focused, symptom-focused, and focused on the possible causes and consequences of dysphoric mood [43]. Each of the 10 items is rated on a Likert scale ranging from 1 (almost never) to 4 (almost always). The total score ranges from 10 to 40. Higher total scores reflect greater self-reported rumination. In the present study we omitted one item of ‘write down what you are thinking about and analyze it’ based on community feedback during the translation and cultural adaptation process as migrant workers thought it was not typical for them to do. The Cronbach’s alpha of RRS in the present study was 0.93, indicating excellent internal reliability.

Perceived social support

The Multi-Dimensional Scale of Perceived Social Support (MSPSS) is a 12-item scale to assess perceived social support [44]. This measure consists of three subscales that examine perceived support from family (four items), friends (four items) and a significant other (four items). Respondents answer on a 7-point scale, from 1 (very strongly disagree) to 7 (very strongly agree). The Cronbach’s alpha in the present study was 0.89, indicating good internal reliability. 

Participant characteristics included self-reported age, years working as a domestic worker in Macao, marital status, education level, type of visa, Cantonese fluency (speaking and understanding), monthly salary, weekly working hours, numbers of days off per month, and residence (i.e., live in or outside of the employer’s house). 

## 3. Data Analysis

### 3.1. Study 1

We computed descriptive statistics for participants’ demographic information. All variables were checked for normality. Pearson correlation was conducted for the relationships between normally distributed variables. Spearman’s rho was used for the relationships between non-normally distributed variables. Item-level missing data for the PSQI was observed for 3 participants. The missing data of PSQI was dealt with using listwise deletion given that less than 5% missingness was observed in the sample [45]. 

#### 3.1.1. Reliability Testing

Internal consistency reliability was assessed using Cronbach’s alpha for the seven PSQI components scores. The values over 0.60 are considered acceptable [46]. Item-to-total correlations (ITC) were calculated to assess the internal homogeneity of the scale. Each component score of PSQI was treated as one separate item. ITC values higher than 0.30 are acceptable [47]. The test-retest reliability was assessed by ICC with baseline PSQI global and component scores and paired 10-day retest scores. A nonparametric bootstrap was used to obtain the 95% confidence interval (CI) of ICC. ICC values higher than 0.75 are considered strong, values from 0.40 to 0.75 are moderate, and values less than 0.40 are considered poor reliability [48].

#### 3.1.2. Validity Testing

Convergent validity refers to associations between two measures that are theoretically related. This was tested with correlations between the PSQI global score and PHQ-9, GAD-7, and RRS. Based on previous literature, we hypothesized that: (a) greater depressive symptom severity would correlate with worse sleep dysfunction [49]; (b) greater anxiety symptom severity would correlate with worse sleep dysfunction [50]; (c) greater level of rumination would correlate with worse sleep dysfunction [51]. Convergent validity was also examined by the associations between the follow-up of PSQI global and component scores and averaged daily sleep parameters from the Actiwatch-2 and sleep diary, separately. We hypothesized that the variables of TST, SL, SE from Actiwatch-2 and daily sleep diary would be significantly associated with PSQI components of ‘sleep duration’, ‘sleep latency’, and ‘habitual sleep efficiency’, respectively. 

Discriminant validity refers to the expected lower association between constructs due to their lack of theoretical relation. This was assessed by correlating the PSQI global score with the ESS, MSPSS and self-reported height. A previous study evidenced poor correlation between ESS and PSQI global [28], this might due to the different goal of ESS, which measures habitual sleepiness rather than actual sleep symptoms [52]. We hypothesized that there would be the negligible correlations between the PSQI global and ESS [28], MSPSS [53] and self-reported height, respectively. 

### 3.2. Study 2

The basic psychometric properties of the PSQI including Cronbach’s alpha, component-to-component correlations (Spearman’s *rho*), and component-to-total correlations (Spearman’s *rho*) were assessed. Construct validity of PSQI was separated into two parts, EFA and CFA. 

The participants were randomly divided into two halves with the RAND formula in Excel. EFA was conducted on the first random sample. Before conducting factor analysis procedures, the suitability of performing factor analysis was assessed based on Bartlett test of sphericity, *p* < 0.001 and the Kaiser-Meyer-Olkin (KMO) of sampling adequacy = 0.64 [54]. EFA was performed using principal component analysis with maximum likelihood estimation to identify the latent factors that explain the common and unique variance of the 19 items of PSQI. An oblimin rotation procedure was conducted. Factors were extracted based on eigenvalues above 1 [55]. The item loading values equal to or greater than 0.3 were retained. 

To verify the factor structure of PSQI, CFA was then conducted on the second random sample to assess the fitness of the structural model based on the identified model obtained in the EFA. The weighted least squares mean and variance adjusted (WLSMV) estimator was used as that the PSQI components scores are ordinal rather than continuous [56]. The adequate goodness of fitness indexes of the model was evaluated and based on standard benchmarks, including the chi-square test of the model (the p value greater than 0.05 would be preferred), comparative fit index (CFI) >= 0.90, Tucker-Lewis index (TLI) >= 0.90, root mean square error of approximation (RMSEA) <= 0.08, and standardized root mean square residual (SRMR) <= 0.08 [57,58]. We also calculated the goodness of fit of other models from the previous studies to make comparisons to other samples. For the best model, the standardized estimated of the factor loading paths was summarized in Figure 1. Descriptive statistics and EFA procedures were conducted with STATA 14.0 (Stata Corp, College Station, TX, US). CFA was conducted using Mplus [59]. All the statistical significance level was set as *p* value < 0.05 with two tails. 

## 4. Results

### 4.1. Study 1

One hundred and thirty-one FDWs with an average age of 39.7 years (SD = 8.3; median = 39; range = 21-59) participated in this study. Their average height was 155.7cm (SD = 6; median = 157.5; range = 130–183). The majority (58.02%) of participants reported to have at least some college or higher educational attainment. The average length as a domestic worker in Macao was 5.1 years (SD = 3.6; median = 4). The average monthly salary was 488.4 (SD = 107.1; median = 480) USD. The reported average weekly working hours were 69.1 (SD = 20.1; median = 70). More than half (59.5%) lived outside of their employer’s home. 

#### 4.1.1. Reliability

The Cronbach’s alpha of the PSQI global scale was 0.63 at baseline (n = 131) and 0.67 at follow-up (n = 61). According to alpha if item deleted analysis, the reliability slightly increased (0.64) either when the component of ‘use of sleeping medication’ or ‘habitual sleep efficiency’ was omitted (see Table 3). The coefficients of item-to-total correlation ranged from 0.37 (‘use of sleeping medication’) to 0.66 (‘subjective sleep quality’ and ‘sleep latency’). The 10-day ICC of the PSQI global score was 0.63. The ICC values of the PSQI component scores ranged from 0.30 (‘use of sleeping medicine’) to 0.58 (‘sleep latency’ and ‘sleep duration’).

#### 4.1.2. Validity

The detailed results of discriminant validity of PSQI are shown in Table 4. The detailed results of convergent validity of PSQI are given in Table 4 and Table 5. No actigraphy variables were found significantly associated with the follow-up of PSQI global and component scores, except inverse correlations between PSQI ‘sleep duration’ and Actiwatch-2 TST (*r_s_* = −0.65, *p* < 0.01) and WB (*r* = −0.43, *p* < 0.01), which show consistency in reporting (higher PSQI scores indicate shorter sleep time). 

### 4.2. Study 2 

#### 4.2.1. Participants Characteristics

Participant characteristics are presented in Table 6. 

#### 4.2.2. EFA Results

The EFA results are displayed in Table 7. Based on the eigenvalue >=1, two factors were obtained, which explained 33.99% and 22.25% variance of data, respectively. Each PSQI component had an acceptable loading, which ranged from 0.38 to 0.77. Five components loaded high on factor 1, which was named ‘perceived sleep quality.’ Two components loaded high on factor 2, which was named ‘sleep efficiency’. This result was the same as that reported by Magee et al. [30].

#### 4.2.3. CFA Results

The two-factor model identified through the result of EFA was tested. We also compared our model with the original one-factor [23] and other two- [28,60] and three-factor models [27,29,61]. Table 8 presented the goodness-fit indices of each PSQI model with second random half of the sample. From the results, the two-factor model based on EFA results presented good fit: CFI = 0.96, TLI = 0.94, RMSEA = 0.065, SRMR = 0.039. However, original one-factor and other two-factor models provided poor fit to the data (see Table 8).

**Table 1 ijerph-17-05219-t001:** The two- or three- structure models of Pittsburgh Sleep Quality Index from the previous studies.

Study (Year) [Reference]	Population	Sample Size	Type of Structure Model	Results	Added Paths (Modifications)
Magee et al. (2008) [30]	Australian adults aged 18 to 59 years old	364	two-factor	Perceived sleep quality (C1, C2, C5, C6, C7); Sleep efficiency (C3, C4).	
Kotronoulas et al. (2011) [28]	Patients with cancer receiving chemotherapy	209	two-factor	Quality of nocturnal sleep: C1, C2, C3, C4, C5; Daily disturbances and management of sleep problems: C6, C7.	
Guo et al. (2016) [51]	Chinese undergraduate students	631	two-factor	Same as Magee et al. (2008).	
Qiu et al. (2016) [62]	US pregnant women	1488	two-factor	Factor1: C1, C2, C3, C4; Factor2: C5, C7.	C2 and C4, C3 and C4
Becker et al. (2017) [63]	Portuguese community-dwelling older adults	204	two-factor	Perceived sleep quality: C1, C2; Sleep efficiency: C3, C4; Daytime function: C5, C7.	
Fontes et al. (2017) [60]	Portuguese breast cancer patients	474	two-factor	Factor1: C1, C2, C3, C4; Factor2: C5, C6, C7.	
Passos et al. (2017) [64]	Brazilian adolescents	309	two-factor	Same as Magee et al. (2008) omitting the sleep medication component	
Otte et al. (2013) [65]	Non-depressed breast cancer survivors	1174	two-factor	Same as Magee et al. (2008).	C1 and C3, C2 and C4, C2 and C6, C2 and C7, C3 and C7.
Gelaye et al. (2014) [61]	Undergraduate students in Chile, Ethiopia, and Thailand	5900	two-factor	Same as Magee et al. (2008).	
Cole et al. (2006) [27]	Community-dwelling adults	417	three-factor	Perceived sleep quality: C1, C2, C6, Sleep efficiency (C3, C4); Daily disturbances (C5, C7).	
Mariman et al. (2012) [66]	Chronic fatigue syndrome patients	413	three-factor	Same as Cole et al. (2006).	
Gelaye et al. (2014) [61]	Undergraduate students in Peru	2581	three-factor	Sleep quality: C1, C5, C7; Sleep efficiency: C3, C4; Other: C2, C6.	
Koh et al. (2015) [29]	Multi-ethnicities population in Singapore	489	three-factor	Perceived sleep quality: C1, C2; Sleep efficiency: C3, C4; Daytime function: C5, C6, C7.	
Otte et al. (2015) [67]	Women with hot flashes	890	three-factor	Same as Cole et al. (2006).	
Zhong et al. (2015) [68]	Peruvian Pregnant Women	642	three-factor	Perceived sleep quality: C1, C2, C5, C7; Sleep efficiency: C3, C4; Sleep medication: C1, C2, C6.	

*Note*: C1 = ‘subjective sleep quality’, C2 = ‘sleep latency’, C3 = ‘sleep duration’, C4 = ‘habitual sleep efficiency’, C5 = ‘sleep disturbances’, C6 = ‘use of sleeping medication’, C7 = ‘daytime dysfunction’.

**Table 2 ijerph-17-05219-t002:** Sleep patterns from actigraphy and daily sleep diary.

Sleep Patterns	Description	Actigraphy	Daily Sleep Diary
Total sleep time (TST)	The total amount of sleep obtained from falling asleep to final awakening (reported in hours and minutes).	√	
The time in bed (TIB)	It was calculated by the interval period between the time to wake up and the reported bedtime the night before.		√
Sleep onset latency (SL)	The time between bedtime and sleep onset (reported in minutes).	√	√
Sleep efficiency (SE)	The result of TST divided by the total time in bed x 100%.	√	√
Wake after sleep onset (WASO)	The number of minutes scored as awake during the sleep period after initial sleep onset.	√	
Number of wake bouts (WB)	The number of contiguous epochs categorized as wake.	√	
Fragmentation index (FI)	The sum of the ‘mobile time (%)’ and the ‘immobile bouts <= 1 min (%)’.	√	
Sleep quality (SQ)	Item from PSQI, ‘During the past month, how would you rate your sleep quality overall?’		√

**Table 3 ijerph-17-05219-t003:** Item characteristics, item-total correlation, alpha if item deleted, 10-day test-retest reliability of PSQI.

Items	Mean	Standard Deviation	Item-To-Total Correlation ^b^	Alpha if Item Deleted	ICC	95% CI of ICC
Component 1: Subjective sleep quality	1.13	0.63	0.66 **	0.55	0.44 **	(−0.07, 0.91)
Component 2: Sleep latency	1.46	0.86	0.66 **	0.56	0.58 **	(0.05, 1.13)
Component 3: Sleep duration	1.37	0.90	0.63 **^,a^	0.58	0.58 **	(0.09, 1.08)
Component 4: Habitual sleep efficiency	0.45	0.74	0.43 **^,a^	0.64	0.35 **	(−0.27, 0.99)
Component 5: Sleep disturbances	1.51	0.61	0.52 **	0.59	0.43 **	(−0.25, 1.10)
Component 6: Use of sleeping medicine	0.27	0.62	0.37 **^,a^	0.64	0.30 **	(−0.18, 0.73)
Component 7: Daytime dysfunction	0.88	0.70	0.59 **	0.57	0.42 **	(−0.21, 1.03)
PSQI-Global (0-21)	7.09	2.86			0.63 **	(0.01, 1.24)

*Note*: Overall Cronbach’s alpha of PSQI is 0.63 at baseline and 0.67 at 10-day retest. * = *p* < 0.05, ** = *p* < 0.01. ^a^ = Spearman’s correlation, ^b^ = Pearson correlation. CI = confidential interval. ICC = intraclass correlation coefficient. Each component score ranges from 0 to 3. Item-to-total correlation means the correlations between each component and the PSQI-Global score.

**Table 4 ijerph-17-05219-t004:** Discriminant and convergent validity of the PSQI at baseline (n = 131).

PSQI Items	PHQ-9 ^a^	GAD-7 ^a^	ESS ^a^	RRS ^a^		MSPSS			Height ^a^
					Total ^a^	Family ^a^	Friends ^a^	Other ^a^	
Component 1	0.27 **	0.26 **	−0.06	0.21 *	−0.23 **	−0.25 **	−0.17	−0.16	0.09
Component 2	0.13	0.06	0.07	0.20 *	−0.21 *	−0.27 **	−0.13	−0.12	0.09
Component 3	0.13	0.10	0.05	0.10	−0.17	−0.18 *	−0.17	−0.07	0.003
Component 4	−0.06	−0.03	0.07	0.12	−0.04	−0.13	0.03	0.07	0.22 *
Component 5	0.38 **	0.3 1**	0.11	0.38 **	.06	0.10	0.03	0.05	0.03
Component 6	0.23 **	0.23 **	0.24 **	0.19 *	−0.05	−0.03	−0.01	−0.09	0.10
Component 7	0.47 **	0.29 **	0.19 *	0.38 **	−0.11	−0.11	−0.19	−0.10	0.12
PSQI-Global	0.36 **	0.28 **	0.17	0.39 **	−0.18 *	−0.21 *	−0.15	−0.07	0.18 *

*Note*: * = *p* < 0.05, ** = *p* < 0.01. ^a^ = Spearman’s correlation. Component 1 = ‘subjective sleep quality’, Component 2 = ‘sleep latency’, Component 3 = ‘sleep duration’, Component 4 = ‘habitual sleep efficiency’, Component 5 = ‘sleep disturbances’, Component 6 = ‘use of sleeping medication’, Component 7 = ‘daytime dysfunction’, and PSQI-Global = the global score of PSQI. PHQ-9 = Patient Health Questionnaire-9. GAD-7 = Generalized Anxiety Disorder-7. ESS = Epworth Sleep Scale. RRS = Ruminative Response Scale. MSPSS = Multi-Dimensional Scale of Perceived Social Support.

**Table 5 ijerph-17-05219-t005:** Convergent validity of PSQI at 10-day retest (n = 61).

Follow-Up PSQI Items	Sleep Diary (n = 58)					Actigraphic Variables (n = 61)					
	TIB ^b^	SL ^a^	TST ^b^	SQ ^a^	SE ^a^	TST ^a^	SL ^a^	SE ^a^	WASO ^a^	WB ^b^	FI ^b^
Component 1	−0.27 *	0.31 *	−0.27 *	0.61 **	0.19	−0.11	0.14	−0.07	0.04	−0.10	0.15
Component 2	−0.19	0.62 **	−0.23	0.44 **	0.34**	−0.09	0.16	−0.16	0.17	−0.07	0.15
Component 3	−0.80 **	0.25	−0.79 **	0.36 **	0.16	−0.65 **	−0.09	0.13	−0.24	−0.43 **	−0.06
Component 4	−0.29 *^,a^	0.15	−0.31 **^,a^	0.30 *	0.19	0.02	−0.01	0.02	−0.05	−0.09 ^a^	0.09 ^a^
Component 5	−0.10	0.28 *	−0.06	0.46 **	0.12	−0.04	−0.09	0.07	0.04	−0.09	0.03
Component 6	/	/	/	/	/	0.15	0.12	−0.18	0.22	0.09 ^a^	0.18 ^a^
Component 7	−0.13	0.30 *	−0.09 **	0.44 **	−0.10	−0.11	0.14	−0.07	0.01	−0.02	0.02
PSQI-Global	−0.48 **	0.48 **	−0.48 **	0.65 **	0.25	−0.23	0.03	0.02	−0.03	−0.21	0.14

*Note*: * = *p* < 0.05, ** = *p* < 0.01. ^a^ = Spearman’s correlation, ^b^ = Pearson correlation. No correlations are calculated between component 6 and sleep diary since no participants reported medical use in the study. Component 1 = ‘subjective sleep quality’, Component 2 = ‘sleep latency’, Component 3 = ‘sleep duration’, Component 4 = ‘habitual sleep efficiency’, Component 5 = ‘sleep disturbances’, Component 6 = ‘use of sleeping medication’, Component 7 = ‘daytime dysfunction’, and PSQI-Global = the global score of PSQI. For the variables of sleep dairy, TIB = ‘the total time spent in bed’, SL = ‘sleep onset latency’, TST = ‘total sleep time’, SQ = ‘self-report sleep quality’, and SE = ‘sleep efficiency’. For the actigraphy variables, TST = ‘total sleep time in bed’, SL = ‘sleep onset latency’, SE = ‘sleep efficiency’, WASO = ‘wake after sleep onset’, WB = ‘wake bouts’, and FI = ‘fragmentation index’.

**Table 6 ijerph-17-05219-t006:** Participant Characteristics for the total sample and random samples used in factor analysis (n = 1363).

Selected Variables	Total Sample (n = 1363)	Sample for EFA (n = 681)	Sample for CFA (n = 682)
Age (years, mean ± SD)	40.99 ± 8.92	40.79 ± 8.87	41.19 ± 8.97
Age group			
18–30	188 (13.79)	92 (13.51)	96 (14.08)
31–40	491 (36.02)	256 (37.59)	235 (34.46)
41–50	482 (35.36)	233 (34.21)	249 (36.51)
≥51	202 (14.82)	100 (14.68)	102 (14.96)
Time working as a domestic worker in Macao (years)	6.07 ± 8.60	6.16 ± 8.86	5.97 ± 8.34
Marital status			
Single, never married	347 (25.46)	175 (25.70)	172 (25.22)
Married	603 (44.24)	302 (44.35)	301 (44.13)
Partnered but not married	97 (7.12)	46 (6.75)	51 (7.48)
Separated	214 (15.70)	100 (14.68)	114 (16.72)
Legally separated	9 (0.66)	6 (.88)	3 (0.44)
Widowed	93 (6.82)	52 (7.64)	41 (6.01)
Education level			
Elementary	24 (1.76)	11 (1.62)	13 (1.91)
High school	489 (35.88)	244 (35.83)	245 (35.92)
Technical/vocational	147 (10.79)	72 (10.57)	75 (11.00)
2-year associate degree	209 (15.33)	107 (15.71)	102 (14.96)
Some college	225 (16.51)	116 (17.03)	109 (15.98)
Bachelor’s degree	265 (19.44)	130 (19.09)	135 (19.79)
Master’s degree or higher	4 (0.29)	1 (0.15)	3 (0.44)
Type of Visa			
Working Visa	1358 (99.63)	678 (99.56)	680 (99.71)
Temporary permanent resident	3 (0.22)	2 (0.29)	1 (0.15)
Permanent resident	2 (0.15)	1 (0.15)	1 (0.15)
Language speaking level (Cantonese)	1.64 ± 1.76	1.63 ± 1.76	1.64 ± 1.75
Language understanding level (Cantonese)	1.72 ± 1.88	1.72 ± 1.91	1.71 ± 1.85
Live-in / live-out			
Live-in	672 (49.30)	333 (48.90)	339 (49.71)
Live-out	691 (50.70)	348 (51.50)	343 (50.29)
Monthly salary (USD)	436.93 ± 101.31	436.72 ± 108.46	437.14 ± 93.70
Working hours per week	65.60 ± 21.99	66.55 ± 22.51	64.66 ± 21.44
Numbers of days off per month	3.71 ± 1.77	3.71 ± 1.13	3.71 ± 1.23

*Note*: EFA = exploratory factor analysis, CFA = confirmatory factor analysis. Cantonese fluency was assessed with a ruler scale, which ranged from the lowest level (0) to the highest level (10). “Live-in/live-out” was asked by “Do you live in your employer’s home?”.

**Table 7 ijerph-17-05219-t007:** The factor loadings in exploratory factor analysis of Pittsburgh Sleep Quality Index (n = 681).

PSQI Items	Factor 1	Factor 2
Component 1: Subjective sleep quality	**0.62**	0.16
Component 2: Sleep latency	**0.48**	0.11
Component 3: Sleep duration	0.13	**0.77**
Component 4: Habitual sleep efficiency	0.00	**0.77**
Component 5: Sleep disturbances	**0.58**	0.03
Component 6: Use of sleeping medicine	**0.38**	0.08
Component 7: Daytime dysfunction	**0.60**	0.05
% Variance explained	33.99%	22.25%

**Table 8 ijerph-17-05219-t008:** Models evaluated for the Pittsburgh Sleep Quality Index and goodness-of-fit indices using confirmatory factor analysis (n = 682).

	Models	χ 2	df	CFI	TLI	RMSEA	SRMR
The second random sample	One-factor	466.20 **	14	0.55	0.33	0.22	0.12
	**Two-factor ^a^**	**50.75 ****	**13**	**0.96**	**0.94**	**0.065**	**0.039**
	Two-factor ^b^	457.24 **	13	0.56	0.29	0.22	0.11
	Two-factor ^c^	464.13 **	13	0.55	0.28	0.23	0.11
	Three-factor ^a^	83.29 **	12	0.93	0.88	0.050	0.093
	Three-factor ^b^	72.42 **	12	0.94	0.90	0.048	0.086
	Three-factor ^c^	83.78 **	12	0.93	0.88	0.084	0.05

*Note*: CFI = comparative fit index, TLI = Tucker-Lewis index, RMSEA = root mean square error of approximation, SRMR = standardized root mean square residual. Two-factor ^a^ model was identified through EFA results. Two-factor ^b^ model is from Kotronoulas et al. (2011) [28]. Two-factor ^c^ model is from Fontes et al. (2017) [60]. Three-factor ^a^ model is from Cole et al. (2006) [27]. Three-factor ^b^ model is from Gelaye et al. (2014) [61]. Three-factor ^c^ model is from Koh et al. (2015) [29].

**Table 9 ijerph-17-05219-t009:** The descriptive and correlation statistic of Pittsburgh Sleep Quality Index (n = 1363).

PSQI Items	1	2	3	4	5	6	7	8	9
1.Component 1: Subjective sleep quality	1								
2. Component 2: Sleep latency	0.41 **	1							
3. Component 3: Sleep duration	0.23 **	0.14 **	1						
4. Component 4: Habitual sleep efficiency	0.08 **	0.04	0.60 **	1					
5. Component 5: Sleep disturbances	0.39 **	0.38 **	0.13 **	0.01	1				
6. Component 6: Use of sleeping medicine	0.16 **	0.04	0.09 **	0.09 **	0.24 **	1			
7. Component 7: Daytime dysfunction	0.39 **	0.28 **	0.16 **	0.06 *	0.38 **	0.26 **	1		
8. PSQI factor 1	0.73 **	0.75 **	NA	NA	0.71 **	0.24 **	0.68 **	1	
9. PSQI factor 2	NA	NA	0.91 **	0.85 **	NA	NA	NA	NA	1
Mean	0.79	1.29	1.29	0.83	1.28	0.27	0.54	3.89	2.12
Standard deviation	0.63	0.83	1.00	1.13	0.67	0.65	0.67	2.04	1.96
Median	1	1	1	0	1	0	0	4	1
IQR	0–1	1–2	1–2	0–1	1–2	0–0	0–1	2–5	1–3

*Note*: * = *p* < 0.05, ** = *p* < 0.01. IQR = Interquartile range. NA = no correlation is presented since items were included in the PSQI factor. Each component score ranges from 0 to 3. All the correlations were Spearman’s *rho* coefficients.

The replicated three-factor model from Gelaye et al. [61] also presented acceptable fit: CFI = 0.94, TLI = 0.90, RMSEA = 0.050, SRMR = 0.093. However, the standardized path coefficient (1.59) between factor 1 ‘perceived sleep quality’ and factor 3 ‘daytime disturbances’ of the model was greater than 1. This result suggested that the factor 1 and factor 3 might have overlapping concepts and should be combined to be one, which was consistent with the EFA identified two-factor model. Figure 1 showed the standardized path coefficients of the two-factor model of PSQI. 

#### 4.2.4. Basic Psychometric Properties of PSQI 

PSQI global scores of 1363 participants ranged from 0 to 17, with the mean score of 6.28 (SD = 3.24). The Cronbach’s alpha of PSQI factor ‘perceived sleep quality’ and ‘sleep efficiency’ was 0.70 and 0.81, respectively. Table 9 provided more detailed information. 

## 5. Discussion

To our knowledge, this is the first study to assess the psychometric properties and the factorial validity of Filipino version of the PSQI. The results demonstrated a low internal consistency of the PSQI global score, but acceptable values for the two PSQI factors. Our literature review revealed a wide arrange of Cronbach alpha of PSQI from 0.57 to 0.89 [24,25,68]. Measures with low alpha may still be useful [69]. The 10-day test-retest ICC values for the PSQI global and component scores demonstrated moderate reliability except for the components of ‘habitual sleep efficiency’ and ‘use of sleeping medicine’, which suggested that sleep is stably assessed using the PSQI global score and some of the component scores within this population. 

Overall, ‘subjective sleep quality’ and ‘sleep latency’ components were most highly correlated with the global score, and components of ‘use of sleeping medicine’ and ‘habitual sleep efficiency’ were least correlated with the global score of PSQI. This pattern of associations suggests that the global score of PSQI reflects ‘subjective sleep quality’ and ‘sleep latency’ more than other components and that ‘use of sleeping medicine’ and ‘habitual sleep efficiency’ are less reliable, consistent with previous studies [62,68]. This is likely due to the infrequent use of sleep medication in this sample (less than 20% reported its use).

The PSQI demonstrated good convergent validity in our sample. Greater sleep dysfunction was significantly associated with higher levels of depression and anxiety, similar to previous research [70]. The PSQI global score and many components were found significantly and moderately associated with RRS. The reason could be explained that these two scales might have conceptional overlap. The RRS assesses respondents’ reflection and brooding on the possible causes and consequences of dysphoric mood [43]. Its association with sleep quality was approved and illustrated among undergraduate students with findings that rumination factors like worry might contribute to cognitive activity, which could affect sleep quality [71].The adequate convergent validity of the PSQI was also supported by the moderate associations between PSQI follow-up and sleep diary variables, which were shown not only on the PSQI global score, but also on other components. We would expect daily assessments of sleep dysfunction to demonstrate higher test-retest reliability than aggregated retrospective reports of sleep problems [72], so the high correlations between daily diary reports and PSQI scores obtained at 10-day follow-up indicate strong reliability for self-reported sleep problems in the sample. In particular, self-reported SQ, SL, and TST were especially highly correlated. This was consistent with previous study findings that sleep patterns of sleep diaries had the high correlations with PSQI items [73].

For actigraphy variables, we only found that longer TST and more WB were significantly associated with longer PSQI ‘sleep duration’. The results were consistent with the previous criterion validity study of PSQI among non-clinical population, which found no significant correlation results between PSQI global and actigraphy variables of TST, SE, WASO, and SL, but significant associations between PSQI sleep duration and TST, as measured by the actigraphy [74]. Similarly, the original PSQI validation study showed a lack of association between the PSQI and PSG with the strongest correlation being *r* = 0.30 between the PSQI and PSG SL [23]. The possible reasons might be that actigraphy or PSG measures actual sleep in real time while the PSQI is retrospective recall measurement, which may hinder accuracy and have reporting biases. Moreover, the low correlations ranging from 0.28 to 0.32 between PSQI components and PSG sleep parameters also supported the difference between objective sleep measures and self-report measures [73].

Discriminant validity was demonstrated by small effect size correlations (<0.30) between the PSQI global score and MSPSS and self-report height. Even some significant associations between PSQI and MSPSS-total and MSPSS-family were found, the associations were still weak. The result was consistent with the previous validation study [53]. 

The present study also examined the factor structure of the Filipino version of PSQI. The EFA identified two factors within the PSQI, which were labeled ‘perceived sleep quality’ for the first factor including the PSQI components of ‘subjective sleep quality,’ ‘sleep latency,’ ‘sleep disturbances,’ and ‘daytime dysfunction’, and the term ‘sleep efficiency’ for the second factor, which including ‘sleep duration’ and ‘habitual sleep efficiency.’ Subsequent CFA evidenced that this two-factor model along with the three-factor model [61] were favored statistically over the original one-factor model [23] and other published two-factor models [28,60]. Although the three-factor model from Gelaye et al. [61] had similar model fit with the our two-factor model, the model fit suggested combining the ‘perceived sleep quality’ and ‘daytime disturbances’ factors. The results and process in the present study were similar with the previous studies [30,61].

The Cronbach’s alpha of PSQI factor ‘perceived sleep quality’ was 0.70, indicating acceptable internal consistency. The Cronbach’s alpha of PSQI factor ‘sleep efficiency’ was 0.81, indicating the good internal consistency. All the components showed high component-total correlations with the PSQI factors, which further supported good internal consistency of the PSQI among FDWs.

Investigators previously argued that a two- or three-factor structure of the PSQI might be a better representation of sleep disturbance than a unidimensional model [27,28,75]. Our study supported a two- factor structure of PSQI, which was consistent with the two-factor model proposed by previous researchers [30,61,64]. Some researchers observed that the removal of ‘use of sleeping medication’ did not have a major impact on the fitness of the CFA models [30]. However, another structure validation study among 309 Brazilian adolescents showed the best two-structure model of PSQI excluding the component of ‘use of sleeping medication’ [64]. In our study, the identified two-factor model fit indecencies improved when this component was removed. Of note, the PSQI global score and cut-off score in defining the poor sleep would be changed when removing ‘use of sleep medication.’ Further studies should explore whether the scale demonstrates incremental validity in assessing sleep dysfunction when ‘use of sleep medication’ component is included in non-clinical samples. 

## 6. Conclusions

Study 1 provided evidence that the Filipino version of PSQI is an adequately reliable and valid assessment instrument useful for quantifying sleep parameters in FDWs. Among the PSQI component scores, the most robust evidence was obtained for ‘subjective sleep quality,’ ‘sleep latency,’ and ‘sleep duration.’ The use of sleep medication is not likely a critical indicator of sleep dysfunction in this population. The findings in Study 2 validated the two-factor structure of the PSQI to assess self-reported subjective sleep disturbance among FDWs. The Filipino version of PSQI scale demonstrated good construct validity. The present study could be referenced for future studies to measure and screen sleep dysfunction among clinical and non-clinical population in the Philippines. 

The current study has some notable strengths. It is the first known study to evaluate the psychometric properties and structural validity of PSQI among Filipino transnational migrants or any Filipino sample. Second, the study design included daily diary self-reported sleep assessments. Third, we used actigraphic assessment as an objective indicator of sleep dysfunction. Despite these strengths, the study has several limitations. First, the sample size only included female domestic workers, limiting generalizability to other transnational migrants and men. This two-factor structure of PSQI may not generalize to all Filipinos or Filipino migrant workers, especially men [27,28,29,30]. Second, participants were recruited using snowball sampling methods in study 1, which is likely to introduce some sampling bias. Third, the factorial validity of the measure could not be assessed given the size of the sample. Further studies that asses a more diverse sample of overseas Filipino workers and evaluate the factorial validity of the Filipino version of the PSQI are needed. Fourth, previous studies used one or several self-reported items instead of the full PSQI scale to measure sleep in epidemiological studies [76,77]. Further studies could explore the utility of a brief version of the PSQI among FDWs due to their very busy schedule. 

## Figures and Tables

**Figure 1 ijerph-17-05219-f001:**
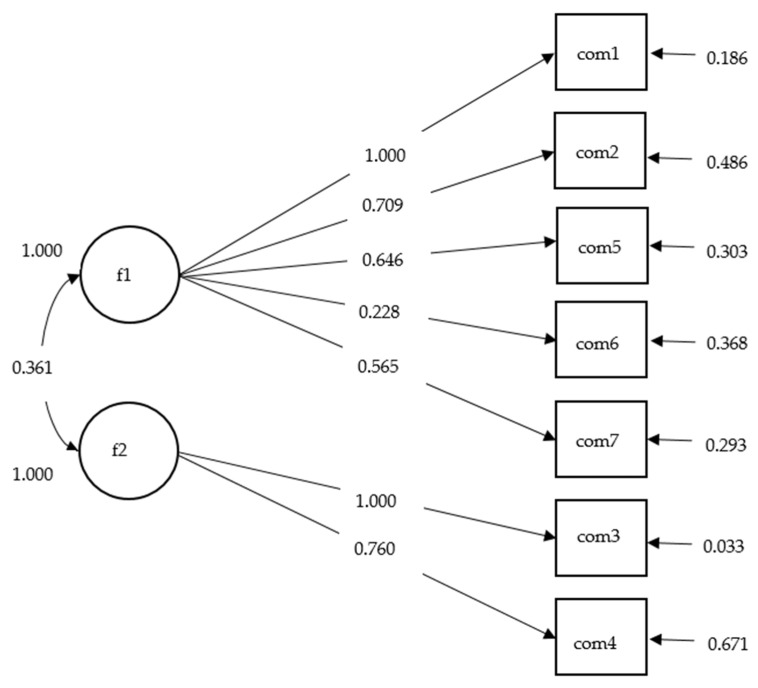
The standardized path coefficients of two-factor model of the Pittsburgh Sleep Quality Index among FDWs in Macao, China (n = 682). *Note*: Factor1 is perceived sleep quality, factor 2 is sleep efficiency. Com1 = ‘subjective sleep quality’, com2 = ‘sleep latency’, com3 = ‘sleep duration’, com4 = ‘habitual sleep efficiency’, com5 = ‘sleep disturbances’, com 6 = ‘use of sleeping medicine’, com7 = ‘daytime dysfunction.

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
