# Peer review of "Psychometric and Structural Validity of the Pittsburgh Sleep Quality Index among Filipino Domestic Workers"

_ijerph, 2020, doi:10.3390/ijerph17145219_

Round 1

Reviewer 1 Report

The study itself fulfills its role in adequacy and methodological merit, by approaching an instrument of great use in current research and by detailing each phase of the process and its respective findings.

However, I point out that my recommendation is conditioned by the high specificity of the sample, which does not allow greater external validity, as well as hindering replication and use in other projects.

I do not discourage the authors, but I believe that submitting this study to a more specific journal on the theme of sleep may be of greater interest.

Author Response

We appreciate the reviewer’s comments and evaluation that the study is worthy of publication.

Reviewer 2 Report

This review pertains to the study entitled “Psychometric and Structural Validity of the Pittsburgh Sleep Quality Index among Filipino Domestic Workers” submitted to the International Journal of Environmental Research and Public Health by Xiong and colleagues. The study examines the psychometric properties of a Filipino version of the Pittsburg Sleep Quality Index (PSQI) and the structural validity in two studies: 1) a study among 131 female Filipino domestic workers (FDWs), and 2) a sample of 1,363 FDWs, of which both were from Macao in the People’s Republic of China. The authors found that the Filipino version of the PSQI had good internal consistency of two factors, and good convergent and divergent validity. This study is overall strong and well written and is worthy of publication. I have a few comments for the authors:

  1. Can the authors describe in greater detail the FDW workforce—specifically the factors that are unique to this group as well as more details about what is known about their sleep quality? I believe this will help in justifying the need for a study in this population. Additionally, can the authors indicate why they are an important group to validate the PSQI in beyond the broader Filipino population? Would results from this study be different in Filipino population with a more diverse vocational composition?
  2. For interpretation of results, it would be helpful if the authors could describe the study setting in greater detail—specifically what role do FDWs play in culture in Macao, and what are their work conditions? Are there any factors that may make their sleep especially different being in China vs. the Philippines, and how might those characteristics influence how they sleep?
  3. The authors state that the two studies used different sampling procedures—Study 1 used snowball sampling, and Study 2 used respondent driven sampling. It may be important for the authors to indicate how these different sampling methods may influence results from both studies, as well as the comparability of the two samples for the purposes of validation of the instrument.
  4. It may be useful to describe in greater detail how the Filipino version of the PSQI was created. It may also be useful to include the instrument as supplementary materials.
  5. The authors noted limited correlations between objective (via actigraphy) and PSQI components. I wonder if the authors could comment on any past literature showing discrepancy between these measures in Filipino culture and describe why this may be the case.

Author Response

Reviewer #2: Comment 2. Can the authors describe in greater detail the FDW workforce—specifically the factors that are unique to this group as well as more details about what is known about their sleep quality? I believe this will help in justifying the need for a study in this population. Additionally, can the authors indicate why they are an important group to validate the PSQI in beyond the broader Filipino population? Would results from this study be different in Filipino population with a more diverse vocational composition?

Author’s reply: We thank reviewer’s comments. We added the following sentences in lines 45-54 to summarize the previous sleep condition studies and the need to validate a sleep scale among migrant workers:

“Migrant workers are likely to experience increased risk of poor sleep and consequent poor health. Migrant workers, especially domestic workers, may be exposed to sleep deprivation due to on-call nature of their work, even during nighttime hours, exposure to traumatic stress, worries about family separation, and long working hours in general  [8-11]. However, the literature on migrant worker sleep problems is scarce. One cross-sectional study about transnational Latino migrant farmworkers revealed that 11% reported daytime sleepiness [9]. In a prevalence study of sleep problems among asylum seekers and refugees, more than half of them (75.5%) reported moderate to severe sleep disturbance [10]. The absence of data on the prevalence of sleep-related problems among transnational migrants is a significant gap in the sleep literature.”

Line 55-60 also reflects the sleep situation among Filipino migrants. Due to the different working environment and culture between domestic workers and international migrant workers, especially among FDWs, we wrote “no previous study has attempted to validate a sleep measure among the large Filipino migrant labor force. Quantifying the burden of sleep impairment requires valid and reliable scales.” in lines 72-74.

Reviewer #2: Comment 3. For interpretation of results, it would be helpful if the authors could describe the study setting in greater detail—specifically what role do FDWs play in culture in Macao, and what are their work conditions? Are there any factors that may make their sleep especially different being in China vs. the Philippines, and how might those characteristics influence how they sleep?

Author’s reply: We thank reviewer’s comments. We added the following sentences in lines 60-68:

“Based on the updated 2018 by the Labour Affairs Bureau, Macao Special Administration Region (SAR), there are 27,348 non-resident domestic workers in Macao. Among them, 52.06% (14,238) come from the Philippines (http://www.dsal.gov.mo). The principal reason for migration from the Philippines is to seek better economic circumstances and improved financial support for their families [15]. Several studies conducted in Macao found a high (>25%) prevalence of anxiety and depression [16]. posttraumatic stress disorder [17], and a 5% prevalence of gambling problems [18], and discrimination was associated with these disorders [19]. All of these disorders can exacerbate sleep problems.”

Reviewer #2: Comment 4. The authors state that the two studies used different sampling procedures—Study 1 used snowball sampling, and Study 2 used respondent driven sampling. It may be important for the authors to indicate how these different sampling methods may influence results from both studies, as well as the comparability of the two samples for the purposes of validation of the instrument.

Author’s reply: We thank reviewer’s comments. The aims of two studies were different. Study 1 aimed to evaluate the reliability, convergent validity and discriminant validity of PSQI. Study 2 aimed to evaluate the construct validity of the PSQI among FDWs. The two samples were comparable. The samples in the two studies were slightly different, which means that they were conducted in different times among different populations. The participants in study 1 were recruited using snowball sampling from February 2016 to July 2016 in Macao (SAR). While, the participants in study 2 were recruited with a respondent driven sampling method in Macao (SAR) from November 2016 to November 2017.

Reviewer #2: Comment 5. It may be useful to describe in greater detail how the Filipino version of the PSQI was created. It may also be useful to include the instrument as supplementary materials.

Author’s reply: We thank reviewer’s comments. We stated the following: “the Filipino versions of the PSQI and Epworth Sleep Scale (ESS) were provided by the Mapi Research Trust (https://eprovide.mapi-trust.org).” in lines 108-109. We did not create the Filipino version of the PSQI.

Reviewer #2: Comment 6. The authors noted limited correlations between objective (via actigraphy) and PSQI components. I wonder if the authors could comment on any past literature showing discrepancy between these measures in Filipino culture and describe why this may be the case.

Author’s reply: We thank reviewer’s comments. The following sentences were added in lines 441-448:

Similarly, the original PSQI validation study showed a lack of association between the PSQI and PSG with the strongest correlation being r = .30 between the PSQI and PSG SL [17]. The possible reasons might be that actigraphy or PSG measures actual sleep in real time while the PSQI is retrospective recall measurement, which may hinder accuracy and have reporting biases. Moreover, the low correlations ranging from .28 to .32 between PSQI components and PSG sleep parameters also supported differences between objective sleep measures and self-report measures [61].

Reviewer 3 Report

Title: Psychometric and structural validity of the Pittsburgh Sleep Quality Index among Filipino domestic workers

The article examines the psychometric properties of the PSQI in samples of Filipino domestic workers. Some issues regarding the methodology of the study need clarification/modification:

Study 1

  1. Sample size determination for study 1 is based on ICC and n=15 is required to achieve the desirable power. Then, why n=131 subjects were recruited in Study 1?
  2. Result of validity: (i) in Table 3, many correlations were not significant and a pattern is evident with components 2-4 as one cluster and the other components form another cluster which needs further discussion. (ii) there were significant correlations of component 4 and PSQI-Global with Height. Any possible explanation and interpretation? (iii) Variables ESS and RSS are included to test for discriminant validity of the PSQI, however, many of their correlations with PSQI global and component scores were significant and the magnitudes are COMPARABLE to those correlations with PHQ-9 and GAD-7. This needs discussion whether the convergent and discriminant validity of the PSQI is supported. (iv) The findings of the low, non-significant correlations with actigraphy variables do not support the hypotheses stated in lines 180-185, the authors then in the discussion stated that such findings are consistent with previous studies. This contradiction needs clarification. (v) Why there is no entry for the correlations of component 6 with sleep diary in Table 4.  

Study 2

  1. EFA: (i) Bartlett’s test and KMO tests should be applied to the EFA sample, instead of the whole sample, as it is the EFA sample submitted to EFA only. (ii) Factor retention in EFA is only based on eigenvalue > 1, which is not recommended as EFA tends to retain more factors as interpretability of factors is also important. Given there were several factors of PSQI (1 to 3 factors) have been reported in the literature, it is desirable to run separate EFAs with number of factors retained from 1 to 3, and then make the decision which one provides a better fit to the data statistically and theoretically.
  2. CFA: (i) it is nice to document the factor structures of PSQI reported in the literature (Table 5), however, the readability of the table could be improved by placing studies reporting the same structure together. (ii) The goodness of fit indexes reported in the text are not the same as those shown in Table 8. (iii) The identification issue of the CFA model is unclear. In CFA, we have to fix some paths in order to make the model to be identifiable. This can be done in 2 ways, either by fixing (a) the variances of the latent variable =1 or (2) one of the indicators of that latent variable = 1. In either case, the corresponding parameter is no longer a parameter to be estimated and hence should not have SE. In Figure 1, the identification method used is not clear – it seems no parameter is fixed for model identification. This guess is further supported by the standardized path coefficient between factor 1 and factor 3 > 1 (1.59), as this parameter should be the correlation between the two latent factors in a CFA. (iv) it is rare to delete an indicator with a significant coefficient in CFA. A standardized coefficient of 0.27 is not small at all, and is not an acceptable reason to delete an indicator of a latent variable. Also, it is not legitimate to compare the goodness of fit indexes across models with different number of indicators, that is, the fit of the two-factor model a* having only 6 observed variables should not be compared with those of the other models having 7 observed variables as fewer indicators will have fewer variations. Anyway, it is unclear what the authors want to recommend: the PSQI should have 6 or 7 components in this target group. (v) Why the two-factor structures reported by Passos et al (2017) and Magee et al. (2008) were not tested with the current sample?
  3. Presentation of tables: replace ‘Component’ with its corresponding name.

Author Response

Reviewer #3: Comment 1. Study 1: Sample size determination for study 1 is based on ICC and n=15 is required to achieve the desirable power. Then, why n=131 subjects were recruited in Study 1?

Author’s reply: We thank reviewer’s comments. Although N = 15 is the minimum sample size based on the hypothesis “ICC value (p0 = 0), the alternative hypothesis pre-specified ICC value (p1 = 0.6), number of replicates (reps = 2), alpha error probability at 0.05, and power at 0.8”. A larger sample size was gathered due to unpredictable factors like participant drop-out rate, and to provide a sufficient sample size for exploring the data in preparation for the larger population-level study we conducted (study 2).

Reviewer #3: Comment 2. Study 1: Result of validity: (i) in Table 3, many correlations were not significant and a pattern is evident with components 2-4 as one cluster and the other components form another cluster which needs further discussion. (ii) there were significant correlations of component 4 and PSQI-Global with Height. Any possible explanation and interpretation? (iii) Variables ESS and RSS are included to test for discriminant validity of the PSQI, however, many of their correlations with PSQI global and component scores were significant and the magnitudes are COMPARABLE to those correlations with PHQ-9 and GAD-7. This needs discussion whether the convergent and discriminant validity of the PSQI is supported. (iv) The findings of the low, non-significant correlations with actigraphy variables do not support the hypotheses stated in lines 180-185, the authors then in the discussion stated that such findings are consistent with previous studies. This contradiction needs clarification. (v) Why there is no entry for the correlations of component 6 with sleep diary in Table 4. 

Author’s reply: We appreciate the reviewer’s comments.

(i) It was discussed in lines 416-423 of the validation parts of discussion:

“Overall, ‘subjective sleep quality’ and ‘sleep latency’ components were most highly correlated with the global score, and components of ‘use of sleeping medicine’ and ‘habitual sleep efficiency’ were least correlated with the global score of PSQI. This pattern of associations suggests that the global score of PSQI reflects ‘subjective sleep quality’ and ‘sleep latency’ more than other components and that ‘use of sleeping medicine’ and ‘habitual sleep efficiency’ are less reliable, consistent with previous studies [61, 63]. This is likely due to the infrequent use of sleep medication in this sample (less than 20% reported its use).”

(ii) It was described in lines 205-207 “Discriminant validity was demonstrated by small effect size correlations (<.30) between the PSQI global score and MSPSS and self-reported height. “

(iii) Lines 452-459 discussed the associations between RRS and PSQI:

“While, the PSQI global score and many components were found significantly and moderately associated with the RRS. The reason could be explained that these two scales might have conceptional overlap. The RRS assesses respondents’ reflection and brooding on the possible causes and consequences of dysphoric mood [35]. Its association with sleep quality was previously demonstrated among undergraduate students with findings that rumination factors like worry might contribute to cognitive activity, which could affect sleep quality [63].”

Line 198 “ESS scores would correlate with the ‘daytime dysfunction’ component from PSQI [44].” was hypothesized to support the convergent validity of PSQI.

(iv) The related literature was added to sentences in lines 441-448.

(v) No entry for the correlations of component 6 with sleep diary in Table 4, it’s because no sleep medication used among these 58 participants.

Reviewer #3: Comment 3. Study 2: EFA: (i) Bartlett’s test and KMO tests should be applied to the EFA sample, instead of the whole sample, as it is the EFA sample submitted to EFA only. (ii) Factor retention in EFA is only based on eigenvalue > 1, which is not recommended as EFA tends to retain more factors as interpretability of factors is also important. Given there were several factors of PSQI (1 to 3 factors) have been reported in the literature, it is desirable to run separate EFAs with number of factors retained from 1 to 3, and then make the decision which one provides a better fit to the data statistically and theoretically.

Author’s reply: We thank reviewer’s comments.

(i) We added “The participants were randomly divided into two halves. EFA was conducted on the first random sample.” in lines 219-220. The KMO score was re-calculated to be 0.64.

(ii)  Yes, we replicated different factor structure models of PSQI in Table 8 and made the decision.

Reviewer #3: Comment 4. Study 2: CFA: (i) it is nice to document the factor structures of PSQI reported in the literature (Table 5), however, the readability of the table could be improved by placing studies reporting the same structure together. (ii) The goodness of fit indexes reported in the text are not the same as those shown in Table 8. (iii) The identification issue of the CFA model is unclear. In CFA, we have to fix some paths in order to make the model to be identifiable. This can be done in 2 ways, either by fixing (a) the variances of the latent variable =1 or (2) one of the indicators of that latent variable = 1. In either case, the corresponding parameter is no longer a parameter to be estimated and hence should not have SE. In Figure 1, the identification method used is not clear – it seems no parameter is fixed for model identification. This guess is further supported by the standardized path coefficient between factor 1 and factor 3 > 1 (1.59), as this parameter should be the correlation between the two latent factors in a CFA. (iv) it is rare to delete an indicator with a significant coefficient in CFA. A standardized coefficient of 0.27 is not small at all, and is not an acceptable reason to delete an indicator of a latent variable. Also, it is not legitimate to compare the goodness of fit indexes across models with different number of indicators, that is, the fit of the two-factor model a* having only 6 observed variables should not be compared with those of the other models having 7 observed variables as fewer indicators will have fewer variations. Anyway, it is unclear what the authors want to recommend: the PSQI should have 6 or 7 components in this target group. (v) Why the two-factor structures reported by Passos et al (2017) and Magee et al. (2008) were not tested with the current sample?

Author’s reply: We thank reviewer’s comments.

(i)We modified and placed studies with same structure together in Table 5.

(ii) We corrected this issue in lines 381-382.

(iii) We fixed the variances of the latent variable to be 1 and modified the figure 1.

(iv) Thanks for your suggestion. We kept the item component 6. Now the final structure model of PSQI includes two factors with all seven PSQI components.

(v) Magee’s model is same with our CFA model in study 2, which comprised of perceived sleep quality and sleep efficiency. Passos’s model is same with Magee’s model, but omitting the sleep medication component. Gelaye’s model is same with Magee’s model.

Reviewer #3: Comment 5. Study 2: Presentation of tables: replace ‘Component’ with its corresponding name.

Author’s reply: We replaced ‘Component’ with its corresponding name in Table 7 and Table 9.

Round 2

Reviewer 1 Report

Apart from my first opinion, I believe that the amendments are sufficient for the manuscript to be published in the journal.

Author Response

Reviewer #1: Comment 1.

Study 1: Apart from my first opinion, I believe that the amendments are sufficient for the manuscript to be published in the journal.

Author’s reply: We appreciate the reviewer’s comments and evaluation that the study is worthy of publication.

Reviewer 3 Report

Title: Psychometric and structural validity of the Pittsburgh Sleep Quality Index among Filipino domestic workers

The article examines the psychometric properties of the PSQI in samples of Filipino domestic workers. While the authors have addressed some of the concerns/problems, there are still some issues need clarification/modification:

Study 1

  1. Please provide the full name of TIB when it is first mentioned (line 119).
  2. The rationale for recruiting 7.7 times more subjects, not just a few more than as planned is not acceptable – One can only conclude that Study 1 was conducted in an unethical way as the study should have been stopped when the target sample size had been achieved.
  3. It is strange to have the same variable (ESS in this case) to test for both convergent and discriminant validity of the same instrument given that the PSQI global includes scores of daytime dysfunction. Very confusing indeed.
  4. The authors are contradicting themselves in the use of RSS for testing the discriminant validity of PSQI as they argued in the Discussion section that the two scales might have conceptual overlap (line 387) – but discriminant validity refers to the expected association between constructs due to their lack of theoretical relation (line 191-192). So, the two constructs conceptually overlapped or lack of theoretical relation?

Study 2

  1. How the two random samples generated from the whole sample (line 202 -203)?
  2. The authors did not clearly indicate that the factor structure of PSQI obtained in EFA was the same as that reported by Magee et al (2008) when they reported their EFA results in lines 316-320, the readers could only find this message by themselves in the Discussion section (line 401) by checking the references provided. This important result should be mentioned earlier and directly given the aim of the study was to test the psychometric properties of the PSQI. Moreover, although I have spent time in matching the previous reported structure of PSQI shown in Table 5 and those being tested in CFA in Table 8, I am still in a mess. For example, it seems that the 2-factor structures reported by Qiu et al (2016) and Becker et al. (2017) and the 3-factor structure reported by Zhong et al (2015) were not tested in CFA.

Author Response

Reviewer #3: Comment 1.

Study 1: Please provide the full name of TIB when it is first mentioned (line 119).

Author’s reply: We thank the reviewer for their comments. The full name “the total time spent in bed” of TIB was added in line 121.

Reviewer #3: Comment 2.

Study 1: The rationale for recruiting 7.7 times more subjects, not just a few more than as planned is not acceptable – One can only conclude that Study 1 was conducted in an unethical way as the study should have been stopped when the target sample size had been achieved.

Author’s reply: We thank the reviewer for their comments. This was the first empirical study ever conducted with migrant domestic workers in Macao. We had several aims: first, we wanted to estimate the burden of objectively assessed sleep problems over time, evaluate the measurement reliability and validity of several scales, gather initial estimates of the expected population prevalence of several disorders, and to evaluate whether the population would be amendable to further research engagement in advance of a planned RDS study (also included in this manuscript). The sample size calculation is really not appropriate for this study, and we have now removed it. We attempted to maximize the utility of the data collection effort, and accomplished several research objectives including the publication of several key studies as a result of this effort. We did not adequately discuss this previously, but in our revision we now state:

“The data of this study was a part of a larger study utilizing actigraphy to determine the burden of sleep dysfunction and related correlates, along with several embedded validation studies [11, 19, 21] and a pilot study for the larger planned respondent driven sampling (RDS) project [20, 31-33].” in lines 96-99.

Reviewer #2: Comment 3.

Study 1:It is strange to have the same variable (ESS in this case) to test for both convergent and discriminant validity of the same instrument given that the PSQI global includes scores of daytime dysfunction. Very confusing indeed.

Author’s reply: We thank the reviewer for their comments. Now, we removed “ESS scores would correlate with the ‘daytime dysfunction’ component from PSQI” from the convergent validity of PQSI in the validity testing section.

Reviewer #3: Comment 4.

Study 1: The authors are contradicting themselves in the use of RSS for testing the discriminant validity of PSQI as they argued in the Discussion section that the two scales might have conceptual overlap (line 387) – but discriminant validity refers to the expected association between constructs due to their lack of theoretical relation (line 191-192). So, the two constructs conceptually overlapped or lack of theoretical relation?

Author’s reply: We thank the reviewer for their comments. Different validation papers used the rumination variable in different ways. Now, we placed the associations between PSQI and rumination variable in the convergent validity section. We moved the discussion part of RRS to lines 360-365.

The validity testing part was modified in lines 178-193:

“Convergent validity refers to associations between two measures that are theoretically related. This was tested with correlations between the PSQI global score and PHQ-9, GAD-7, and RRS. Based on previous literature, we hypothesized that: a) greater depressive symptom severity would correlate with worse sleep dysfunction [47]; b) greater anxiety symptom severity would correlate with worse sleep dysfunction [48]; c) greater level of rumination would correlate with worse sleep dysfunction [49]. Convergent validity was also examined by the associations between the follow-up of PSQI global and component scores and averaged daily sleep parameters from the Actiwatch-2 and sleep diary, separately. We hypothesized that the variables of TST, SL, SE from Actiwatch-2 and daily sleep diary would be significantly associated with PSQI components of ‘sleep duration’, ‘sleep latency’, and ‘habitual sleep efficiency’, respectively.

Discriminant validity refers to the expected lower association between constructs due to their lack of theoretical relation. This was assessed by correlating the PSQI global score with the ESS, MSPSS and self-reported height. A previous study evidenced poor correlation between ESS and PSQI global [28], this might due to the different goal of ESS, which measures habitual sleepiness rather than actual sleep symptoms [50]. We hypothesized that there would be the negligible correlations between the PSQI global and ESS [28], MSPSS [51] and self-reported height, respectively. “

Reviewer #3: Comment 5.

Study 2: How the two random samples generated from the whole sample (line 202 -203)?

Author’s reply: We thank the reviewer for their comments. We modified the sentence to read “The participants were randomly divided into two halves with the RAND formula in Excel.” in line 199.

Reviewer #3: Comment 6.

Study 2: The authors did not clearly indicate that the factor structure of PSQI obtained in EFA was the same as that reported by Magee et al (2008) when they reported their EFA results in lines 316-320, the readers could only find this message by themselves in the Discussion section (line 401) by checking the references provided. This important result should be mentioned earlier and directly given the aim of the study was to test the psychometric properties of the PSQI. Moreover, although I have spent time in matching the previous reported structure of PSQI shown in Table 5 and those being tested in CFA in Table 8, I am still in a mess. For example, it seems that the 2-factor structures reported by Qiu et al (2016) and Becker et al. (2017) and the 3-factor structure reported by Zhong et al (2015) were not tested in CFA.

Author’s reply: We thank reviewer’s comments. We added “This result was the same as that reported by Magee et al [30].” in line 316.

We did not test the 2-factor structures reported by Qiu et al (2016) and Becker et al (2017), because both of these two studies did not put component 6 ‘use of sleeping medicine’ in the structure model. We did not test the 3-factor structure reported by Zhong et al (2015), because this study used the same component repeatly in different factors, specifically, the factor ‘perceived sleep quality’ included C1, C2, C5, and C7, factor ‘sleep efficiency’ included C3 and C4, factor ‘sleep medication’ included C1, C2, and C6. This was described in detail in Table 5.